# The Use of Mechanical Ventilation Support at the End of Life in Motor Neurone Disease/Amyotrophic Lateral Sclerosis: A Scoping Review

**DOI:** 10.3390/brainsci12091162

**Published:** 2022-08-30

**Authors:** Eleanor Wilson, Jeong-Su Lee, David Wenzel, Christina Faull

**Affiliations:** 1Nottingham Centre for the Advancement of Research in Palliative and End of Life Care (NCARE), School of Health Sciences, University of Nottingham, Nottingham NG7 2AH, UK; 2Guy’s and St. Thomas NHS Foundation Trust, London SE1 7EH, UK; 3Health Sciences, University of Leicester, Leicester LE1 7RH, UK; 4LOROS Hospice, Leicester LE3 9QE, UK

**Keywords:** motor neurone disease/amyotrophic lateral sclerosis, mechanical ventilation, end of life, dying, scoping review, withdrawal of ventilation

## Abstract

There has been an increase in the use of mechanical ventilation (MV) in motor neurone disease (MND) to alleviate symptoms related to hypoventilation. Little is known about its use at the end of life, and the withdrawal of MV is a challenging topic. A scoping review was conducted to map the existing evidence of ventilation use at the end of life in MND. The Joanna Briggs Institute methodological framework was used, and a total of 31 documents were included in the data extraction. Boarder themes around place, planning, cause and comments on death were identified. Our findings show that the focus of the literature has been on the process of the withdrawal of ventilation, creating the subthemes of timing and reason for withdrawal, along with ethical, emotional, and practical issues, medications used and the time until death. There is a foundation of evidence to guide the process and discuss the ethical and emotional issues of withdrawing ventilation. However, there remains limited evidence from patient and family member perspectives. Importantly, there is almost no evidence exploring dying with ventilation in place, the active or passive decisions around this process, how the interface might impact the dying process, or what families think about this.

## 1. Introduction

Motor neurone disease (MND) incorporates several neurological conditions, where the degeneration of the motor nerves causes the muscles to progressively weaken. Amyotrophic lateral sclerosis (ALS) is the most common form, and the terms are often used interchangeably. In the UK, the term MND is predominantly used and will be used throughout this review. Over time, people with MND lose their ability to move, speak, breathe and communicate. Mechanical ventilation, using a close fitting nose-mask or facemask (non-invasive ventilation: NIV) or a tube in the neck (tracheostomy/invasive ventilation: IV), can help to support breathing and may increase survival [1,2,3]. At first, this treatment may be used overnight to relieve symptoms related to hypoventilation. As the respiratory muscles weaken, it may be necessary to use the ventilator more, until respiratory functions cannot be sustained without it.

Some patients will have ventilation initiated in a respiratory crisis and may be unable to be subsequently weaned off it [4]. In the UK, approximately 1% of those with MND receive IV, and current evidence shows that 81% of those cases were treated via emergency tracheostomy [5]. Evidence from Norway, however, shows that 28% of people with MND using ventilation underwent tracheostomy (72% used NIV) [6,7]. Across the world, rates of the use of both forms of ventilation vary. Spittel et al.’s study showed that the overall proportion of German MND patients using ventilation therapy was 28% [8]. This is in line with worldwide statistics, which vary from higher rates in Italy of approximately 46% to Taiwan, where rates are lower, at around 17% [8]. As other aspects of the neurological condition worsen, such patients may become unable to communicate other than via eye movements (locked-in syndrome) [9,10] or may even lose all ability to communicate (a total locked-in state) [11].

As their illness progresses, some people may wish to decide whether or not they want to stop ventilation [12]. Whilst there is some tentative evidence that suggests the majority of patients who are dependent on ventilation wish to continue to use it until their death [13], a smaller proportion wish to stop their ventilation sooner [14]. Stopping, or withdrawing, ventilation at this stage means that the patient will die, and most likely within a very short space of time [15,16,17,18]. There is currently little published evidence about what it is like for patients using mechanical ventilation to die, or to have ventilation withdrawn, what informs this decision, or how patients and families are supported throughout the experience [19]. Yet, the use of mechanical ventilation across Europe and North America is increasing in a range of conditions, including chronic obstructive pulmonary disease (COPD), obesity-related respiratory failure, neuromuscular diseases, including motor neurone disease [20], and, presently, COVID-19-related respiratory disease [21,22].

Legally, in the UK and many other countries, ventilation is considered a medical treatment and the refusal of such treatment by a patient who has capacity must be complied with, and to continue that treatment without their consent constitutes a criminal offence [23,24,25]. However, this is not the case in all countries, with Japan [26] and Israel [27] being notable exceptions. The ability to control the potential future discontinuation of ventilation may be a factor for some patients in making decisions about starting this treatment [12,28]. In the UK, evidence suggests that doctors and nurses lack the knowledge and understanding of the processes and its legal issues [29]. Professional guidance publications note the general lack of published research evidence in this field and the need for further exploration of the issues surrounding it [25]. Little is known about the extent or nature of the discussions between patients, families and healthcare professionals, how patient choice is enabled or how decisions about the use of ventilation at the end of life are subsequently actioned. This scoping review maps the nature of the evidence of the use of ventilation at the end of life in MND [30].

## 2. Methods

A scoping review specifically aims to identify the scope of the current literature and gaps in knowledge and to clarify concepts. They are useful in examining emerging evidence and can report on the types of evidence available [31]. They differ from systematic reviews in that they do not attempt to address the feasibility, efficacy or appropriateness of a particular treatment or practice [31], but can be used to identify research priorities, contextual information and background information on various phenomena or concepts [32]. This scoping review, based on methodological guidance from the Joanna Briggs Institute (JBI) [33], draws together relevant international literature to identify the types of evidence available, to map and summarise the evidence in order to establish what is known and to identify the gaps in knowledge about the use of ventilation at the end of life in MND [31]. The protocol, and the current paper reporting this scoping review, are in line with the Preferred Reporting Items for Systemic Reviews and Meta-Analyses Extension for Scoping Reviews (PRISMA-ScR). The protocol was registered prospectively with the Open Science Framework (https://osf.io/) on 14 October 2021 [34].

The JBI methodological framework for conducting scoping reviews was drawn upon to identify the research aims, inclusion criteria, search strategy, selection of studies and the data extraction process of charting, collating and presenting the results of this paper [33]. The inclusion criteria and types of evidence sources are set out in Table 1 below. Given the need for a scoping review to have a broad scope, the use of the PPC framework is recommended for developing the focus of the scoping review [32,33]. The PPC identifies:

**Population**—people with MND and the health care professionals and families involved in their care;

**Concept**—the use of mechanical ventilation at the end of life, whether non-invasive ventilation (NIV) or invasive/tracheostomy ventilation (IV);

**Context**—healthcare or home settings;

**Aim:** to establish what is known about the use of ventilation at the end of life in MND.

## 3. Search Strategy

### 3.1. Inclusion/Exclusion Criteria

Publications were included if the full text could be obtained from any of the University of Nottingham libraries, King’s College libraries or e-libraries, free online open access resources or legal deposit libraries, or by directly contacting the authors. In order to be as inclusive as possible, a small amendment was made to the protocol, as it became apparent that some opinion-based articles provided valuable clinical expertise and should not be excluded on this dimension alone. A further revision to include articles not published in English, where they could be translated, was also made.

### 3.2. Search Sources

The search limitations levied were ‘humans’ and ‘all adults (19 years plus)’. No date restrictions were imposed, and the search terms can be found in Box 1. For pragmatic purposes, and in order to manage the amount of initial citations, searches were carried in three segments. First, AMED, Embase, Ovid MEDLINE^®^ and Journals@Ovid Full Text were searched via the Ovid online search engine. To try to ensure that no articles were missed, the Pubmed Central and Cochrane databases were then also searched using slightly amended search terms appropriate for the individual database. Finally, EBSCOhost was used to search CINAHL. Once the main searches had been conducted, searches of Google Scholar, MedRxiv pre-print service and reference lists were also undertaken. Once duplicates were removed, a total of 5120 citations were identified.

Box 1Search terms.MND/or motor neurone disease/or motor neuron disease/or ALS/or amyotrophic lateral sclerosis/or ‘Lou Gehrig’s disease’Andventilation/HMV/“home mechanical ventilation”/or NIV/or non-invasive ventilation/or non-invasive/or TV/or tracheostomy ventilation/or IV/or invasive ventilationAnd“end of life*” or end-of-life or end of life care/or palliative* or “palliative care”/or withdraw*/or dying/or die/or mask remov*And“decision making”/or decision* or “decision mak*” or decision-mak* or decid* or choice* or communicat* or discuss* or conversation* or “advance decisions*”/or ADs/or “advance care plan*”/or ACP/or adherence

### 3.3. Screening and Study Selection

Titles were reviewed by E.W., resulting in 175 documents for abstract review by E.W. and J.-S.L. The abstract review was undertaken independently and then the reviews were compared until an agreement was reached. Documents were excluded if they did not focus sufficiently on MND, ventilation and end of life. Also excluded were letters, responses and conference abstracts. Where it remained unclear whether a document should be included, these cases were kept for the review of the full text. This phase resulted in 71 documents for the full text review. These 71 documents were independently reviewed by E.W., J.-S.L. and D.W., and notes made were collected in a shared file to confer opinions. Where disagreement occurred, documents were re-reviewed and discussed between E.W., C.F. and J.-S.L. until agreement was reached. Seven further papers were identified via other methods (see Figure 1), four of which were included. Documents excluded were primarily omitted because they did not provide enough focus on ventilation or the end of life. Despite no date restrictions being imposed, one paper from 1986 was excluded, as it was felt that this precluded a number of relevant developments in home mechanical ventilation from being considered, substantially limiting the utility of the article for the purposes of this review. Articles were also excluded at this stage if it was not possible to distinguish their implications for MND patients, amongst other conditions, or if the focus was entirely on ethical or legal debates.

After the final agreement, 31 documents were included for extraction and charting [35,36]. The team included the Guidance for Professionals from the Association of Palliative Medicine of Great Britain and Ireland in their analysis [22] (n = 31). This was identified as a key document, as it provides an overview of the issues, draws on largely unpublished and primary evidence from health professionals and bereaved family members and provides detailed guidance for professionals involved in the withdrawal of assisted ventilation. Given the nature of the topic, the size of the population and the methodological limitations imposed by researching the process of dying, detailed case presentations and review articles based on clinical experience are an important source of knowledge in this field and were therefore essential to include in this scoping review. A PRISMA flow chart of the literature screening process is shown in Figure 1. EndnoteX9© (Clarivate, Philadelphia, PA, USA) was used throughout the process to sort, manage and store the articles.

### 3.4. Data Extraction and Charting

Critical appraisal is not an essential part of a scoping review and was not undertaken [33]. However, a charting process was carried out to identify key areas of information relevant to the aim of the scoping review. Once the included papers had been read and the data had been extracted to create a table of study characteristics (see Appendix A), a data charting form was used to map the findings across the papers [36], as shown in Appendix A. These key areas of information were charted and assessed for their contribution to the aim of the review, based on their focus and content. Using a deductive approach, the relevant areas of interest were charted and then developed into themes by grouping key issues and providing overarching descriptors. These were discussed and considered by E.W. and C.F. and reviewed by J.-S.L. and D.W. Five themes were identified: place of death, cause of death, comment on death, planning and ventilation withdrawal. Within the theme of ventilation withdrawal, several important subthemes were identified: timing and reason for withdrawal, ethical and emotional issues, practical issues, inclusion of medication use, time until death and who should be involved in the process.

## 4. Findings

The review identified 31 documents (see Appendix A) from those 19 papers reporting empirical research, of which 11 used quantitative methods [11,37,38,39,40,41,42,43,44,45,46] and 8 used qualitative methods [13,29,47,48,49,50,51,52]. The documents which did not report empirical research studies presented discussions based on clinical case examples (n = 6) [17,18,53,54,55,56] or provided reviews or opinion pieces (n = 4) [57,58,59,60]. One literature review [61] and one set of clinical guidance recommendations for the withdrawal of ventilation were identified [25].

### 4.1. Type of Study

Of the 19 empirical papers included, a range of study designs and methods were reported (see Figure 2). These were primarily interview studies (n = 10) [13,29,37,38,47,48,49,50,51,52]. Two used a more structured approach to the interviews [37,38]. The study by Moss et al. [37] used structured interviews with questionnaires to identify patients’ functional abilities and demographic indexes, mode of ventilation, attitudes to life and their decisions about starting and circumstances for wishing to stop ventilation (included as the interview and questionnaire methods in Figure 2).

Of the 10 studies employing interview methods, 4 had a focus on patient and/or family caregiver perspectives [37,38,47,50], 3 had a focus on the views of healthcare professionals [29,48,49] and 3 reported findings from both healthcare professionals and family caregivers [13,51,52].

Along with the study by Moss [37] noted above, three further articles reported a survey or questionnaire designed to determine the views of healthcare professionals [41,45,46]. Three cohort studies were also included [39,43,44], along with three studies where reviews of autopsy or medical records had been used [11,40,42].

Figure 3 shows the numbers of documents plotted by their country of origin. Two studies were conducted in more than one country. Escarrabill et al. [40] conducted data collection from five hospitals from the north of Italy and Spain, and Thurn et al.’s [45] online survey gathered data from physicians involved in MND care in Belgium, Germany, Ireland, Italy and the UK. Given the prevalence of both NIV and IV use in Japan, we recognised that there may be some non-English literature that we were unable to include, resulting in only two papers that met the inclusion criteria for this review. The paper by Meyer et al. [39] was originally published in German. Google translate was used to gain a clear translation of the study to enable us to include this paper.

### 4.2. Type of Ventilation

Findings show (see Figure 4) that 11 papers focused on NIV [13,18,38,42,44,49,52,54,57,58,59], 7 focused on IV [11,17,45,53,55,56,60] and a further 13 included both [25,29,37,39,40,41,43,46,47,48,50,51,61]. However, those patients on ventilation were often only part of the population included, and for those papers covering both forms of ventilation, it was not always possible to distinguish these patients in the findings.

## 5. Thematic Evidence

### 5.1. Place of Death

There are some comments on the place of death of people using ventilation support across the papers but little exploration of what may influence this. Findings show variability in the proportions of hospital and home deaths reported [40] (examples can be seen in Table 2), unless ventilation was withdrawn, as demonstrated by cases in Dreyer’s Danish study, which indicate that patients are more likely to die at home [11]. In Ushikubo’s study [48], the place of death was linked to the time when patients started using home care services, with those dying in hospital more likely to have started using services later. The author also found that the place of death was related to whether or not the home care nurse recognised the dying phase [48]. Escarrabill’s study of patients across Italy and Spain showed that the place of death was significantly related to the characteristics of ventilation during the last week of life. Those using NIV with facial masks were more likely to die in hospital, and the frequency of using a volume ventilator was higher in those that died at home [40].

### 5.2. Cause of Death

Cause of death was recorded by a small number of the included studies [48,49,50]. Of note, Veronese et al. reported that 20% of patients died suddenly [50], and in Ushikubo’s study, a small proportion of deaths were reportedly caused by ‘accident’ (machine failure or dislocation of the mask) and ‘caregiver ability’ (forgetting to turn the ventilator back on) [49]. Burkhardt et al. [42] provided more details on the cause of death for those using NIV in their autopsy study of 80 consecutive patients dying with MND, of whom 38 had used mechanical ventilation. A comparative analysis indicated that bronchopneumonia was the only cause of death that occurred more frequently with NIV users [42]. Across all the patients with MND, 72/80 died of respiratory failure, primarily pneumonia, caused by either aspiration or by bronchopneumonia, followed by hypoxia and then combined causes. A further six died as a consequence of assisted suicide, one died of complications after PEG insertion and one died from cardiac disease [42].

### 5.3. Comment on Death

Across all papers, there was limited reference to, or comment on, the manner of death. There were some references to this being ‘peaceful’ [38,47] but no exploration of what this term might mean and little to help us understand what dying with ventilation in place, in particular, might appear to be like. Baxter et al. [13] provided some insight from their interviews with family caregivers of patients with NIV, culminating in a small section headed ‘Peaceful final moments’, which referred to ‘changes in breathing’ and the person looking as though they had ‘gone to sleep’. For those reporting on the withdrawal of ventilation, there is some further detail, often referring to the use of symptom management and deep sedation to indicate ‘peace’ at the time of death [11,46,61]. Where signs of agitation, alertness or movement were identified during the dying phase, these prompted clinicians to administer medications [43,46]. In order to illustrate this transitional period in ventilation withdrawal, Faull and Wenzel collated clinician responses to questions about why medication was administered after ventilation was withdrawn. Examples included:‘*Slight distress and movement*’;‘*Although thought to probably be comfortable, he had subtle flickering eyelids–not clear if it represented reflex/involuntary*’;‘*Looked unsettled and family started to get anxious that he was distressed*’;‘*Laboured breathing, although not obviously distressed*’ [46] (Table 2, p3).

LeBon and Fisher’s descriptive case study of withdrawing long-term invasive ventilation offers some insight, reporting:

‘...*he maintained shallow spontaneous respiratory movement but remained comfortable and no additional opioid or sedative was required. He was certified dead 15 min later*’.[55] (p263)

Messer at al. offered a discussion of the physiological changes drawn from their six case studies of patients using NIV [18]. These included an increased respiratory rate, despite a lack of consciousness, and the opening of eyes at the moment of death. Both were interpreted as physiological reflexes rather than patient distress [18]. Only one paper provides comments on turning off the machine after death, when the person has died with ventilation in place [13]. The findings presented show that, once the patient has died, leaving the machine in place, making noise and being perceived as continuing the breath for the person could cause distress:

“*One of the difficulties afterwards was–is he still breathing, because the machine was breathing for him and she used her judgement to make the decision to turn the machine off because that would be a very distressing situation where the machine was breathing for somebody who had passed away*”.[13] (HCP6, p520)

Baxter et al.’s findings illustrate the need for a clear understanding of the ways that ventilation functions when applied by both family members and health care professionals, so that there is no confusion about whether or not the patient has died [13]. Guidance for the management of this circumstance was subsequently included in the UK guidance [25].

### 5.4. Planning

A large proportion of the articles included references to advance planning (23/31). This theme of advance planning included mentions of Advance Decisions to Refuse Treatment and Do Not Attempt Resuscitation forms, expressed wishes recorded in medical notes, advance directives, living wills and discussions about end of life care [11,13,17,18,25,29,37,39,41,44,45,46,48,49,51,52,53,55,56,57,58,59,60,61]. The study by Moss et al. suggested that the uptake of some form of advance planning is high for those with MND, with 79% of the 50 patients using ventilation in their study having left directives in place. However, patients were more likely to express such preferences to family members rather than clinicians [37]. The articles drawing on case presentations were able to illustrate the benefits of discussions about ongoing ventilation use prior to the dying phase in practice [17,18,53,55,56]. Each of the opinion- and review-focused articles also referred to the importance of advance care planning and/or decisions made in advance of the elective withdrawal of ventilation [56,57,58,59,60,61]. The UK guidance provides clear direction on the deciding and planning of withdrawal [25]. All recommendations suggest that elective withdrawal be discussed with the patient, their family and the healthcare team. Indeed, Faull and Oliver [58] referred to the UK guidance, which recommends that, prior to commencing ventilation, there be an explanation provided to the patient that it can be stopped at any time, and that there should be ongoing opportunities for the patient to discuss their wishes for its continuing or withdrawal [58]. The guidance states:

“*A senior doctor should take responsibility for validating the decision to withdraw assisted ventilation and the planning and undertaking of the withdrawal. ...A senior doctor needs to ensure that it is a settled decision of a patient with capacity or that the advance decision is valid and applicable*”.[25] (p15)

Turner at al. noted that choice, and being able to change their mind, is essential for people with MND, making it crucial that individual’s wishes are discussed. They state:

“*Advance care planning is not a single event...Individual preference may change over time and discussion enables people to develop a more considered view concerning assisted ventilation and resuscitation*”.[60] (p471)

LeBon and Fisher illustrated this change in patient preferences over time and the role of such an advance care plan in relation to their presented case study of a patient with invasive ventilation [55]. However, findings from Chapman et al.’s [52] study show that these conversations do not always take place. Despite recognising the initiation of NIV as an accepted ‘trigger point’, clinicians in the study cited several challenges for communication about the end of life, including when to have the conversations, having the time to have this discussion properly and trying to ‘assess patient/family readiness’ for such communications [52]. Clinicians also reported that having multiple teams involved in the care could result in contradictory information and confusion about who was responsible for these conversations. Bereaved family members cited that clinicians should initiate such conversations and highlighted that not discussing withdrawal or being given time to plan and prepare could have lasting impacts [52].

### 5.5. Withdrawal

The withdrawal of ventilation was a key overarching theme identified by the extraction and charting process. Twenty articles, along with the UK guidance [25], addressed this theme in enough detail to contribute to the aim of the review [11,17,18,25,29,39,41,43,45,46,47,51,53,54,55,56,58,59,60,61].

#### 5.5.1. Timing and Reason for Withdrawal

Reasons for withdrawal and the timing of the process were interlinked. As Berger narrated in his presented case, the patient was aware of his diminishing abilities to communicate and therefore wished to have his ventilation withdrawn before he lost all avenues of communication and became totally locked in [53]. Loss of all communication was cited as a central reason for withdrawal [11,17,55,56]. Findings from Thurn et. al.’s [45] study of German physicians’ attitudes toward end of life decisions in MND showed that respondents were more likely to accept psycho-existential concerns as reasons for withdrawing ventilation than physical concerns. Overall, participants in this study assumed a more reactive role, being more in favour of undertaking withdrawal at the patients’ request rather than offering this option [45].

Dreyer et al.’s [11] study found that all 12 cases wanted to withdraw ventilation before developing a totally locked-in state. The reasons given was generally a loss of will to live, but in some cases more specific issues, such as recurrent infections, inability to speak, and bleeding from cancer were reported [11]. Messer et al. provided details about the patient’s Advance Decisions to Refuse Treatment (ADRT), with patients directing withdrawal when ‘prolonged unconsciousness due to MND occurred’, and when there was further deterioration of the ‘bulbar function or mobility’ [18]. Messer et al. [18] contributed the only article to provide comment on timing the withdrawal beyond the UK guidance [25]. They suggested that undertaking withdrawal in the evening can be easier in terms of workload for the staff involved but may be challenging for community staff and with respect to access to medications.

#### 5.5.2. Ethical/Emotional Issues of Withdrawal

A small number of key papers illustrate the complex emotional and ethical issues that can be associated with the withdrawal of ventilation [17,29,41,51,54,55]. Two papers from a study led by Faull [41,51] highlight that, for both families and healthcare professionals, the issues of ethics, morals and legality are interlinked. While healthcare professional may be clear on the ethical and legal theories, their application to practice was more complex and could be personally burdensome [41]. As Faull et al. stated:

*‘While the ethical logic is understood, the process of NIV withdrawal, for some at least, feel different to the withdrawal of other treatments’*.[41] (p46, emphasis on original)

Gaining consensus not only with the family, but also with colleagues, was considered important to support the process [17,29,41,54,55]. The negative impacts of health care professionals refusing to participate in the withdrawal process could cause delays, as could patients not being able to die in the place of their choosing and unfamiliar staff being involved, which could impact on the quality of the end of life care delivered, such as the control of symptoms [51,54]. Considerable distress could also be caused to patients and families when they were aware that the decision was not supported by the whole care team [51]. The rarity of this event meant that drawing on the expertise of more experienced colleagues was essential [51]. The involvement of the multidisciplinary team was highlighted as important [17,51,55,60]. Turner et al. broadly suggested that this would involve respiratory and palliative medicine colleagues but would be unlikely to involve neurology [60]. More specific examples of whom might be involved in the process are found in the articles providing case examples, such as Gleeson and Johnson [17] and LeBon and Fisher [55]. Even when health care professionals supported the decision and felt privileged to be able to help the patient carry out their wishes, participants still found the process emotionally and morally challenging [29,46,51,58].

#### 5.5.3. Practical Issues/Medications Used in Withdrawal

There were also a number of practical issues raised in the documents that focused on withdrawal, and these are drawn together in the UK guidance [25]. Alongside the guidance, some articles noted the considerable time needed to plan the process and communicate with all those involved [18,41]. These practical issues were linked to the complexity of getting support and agreement from the multi-disciplinary team. Other practical issues raised focused on the use of medications, both pre-emptively and as methods of symptom control, during the process. While there was a recognition that medication dosages will be individually tailored, morphine, diazepam, midazolam and levomepromazine were reported to be commonly used as background infusions and to manage breakthrough symptoms during the process of withdrawal (for examples, see [11,18,46,53,55]). Kettemann et al. [43] compared ‘continuous deep sedation’ for those without any ventilator-free tolerance with ‘augmented symptom control’, where sedation was not intended for those with some ventilation-free tolerance (also see [39,46]). Kettemann et al.’s discussion concluded that augmented symptom control may be seen as closer to the natural dying process, yet continuous deep sedation, though potentially more invasive, may be seen as less stressful and fearful for the patient. However, such continuous deep sedation may feed into anxieties that healthcare professionals may have ‘caused’ the death through their actions [43]. Making provision for anticipatory symptom management prior to the interface removal was predominantly considered optimal, as additional doses of medication could then be used to manage any further symptoms that might indicate discomfort or distress [18,39,46,53,61]. Messer at al. offered some valuable comments, suggesting the involvement of additional senior clinicians in order to share the responsibility for dosages and assessment of responses to such medications. They also recommend the use of a separate room to prepare medications away from the patient and family, for both safety and privacy [18].

#### 5.5.4. Time until Death

The time from the interface removal until death varied considerably. Where deep sedation was used for those with no ventilator-free tolerance, an expected correlation between the use of deep sedation and a shorter time until death was identified [39,43]. On average, the time from withdrawal to death was cited in the range from ‘moments’ to 33 h [18,39,43]. Faull and Wenzel noted, in their study based on large (n = 46) prospectively collected database, that for those (n = 4) with tracheostomy ventilation, death occurred within 30 min of ventilation removal, but for those with NIV (n = 42), there was considerable variation, ranging from ‘moments’ to 54 h, with a median of 30 min [46]. The authors reported that the whole process, from the decision to actively start the withdrawal to the patient’s death, took on average 5 h, and for 60% of patients this was 2 h or less. However, for 10%, the process was much longer, notably beyond a working day [46]. Individual case study examples can be found in the articles by Berger [53], Gleeson and Johnson [17], LeBon and Fisher [55] and within the UK guidance [25].

## 6. Discussion

The literature on the withdrawal of home mechanical ventilation in MND is growing, with greater detail and consensus being generated. The level of the patient’s needs and the complexity of the symptom management require healthcare professionals to be present at the death and share their experiences and knowledge of these events. Indeed, there are a number of case study articles and empirical studies cited in this review that offer insight into the processes and provide guidance for other clinicians. In 2015, this evidence, including largely unpublished primary evidence from health professionals and bereaved family members, was drawn together to inform the UK national guidance on the withdrawal of ventilation [25,52,62]. While our knowledge of withdrawing ventilation at the patient’s request is increasing, this scoping review has identified a significant gap in our understanding of what happens when someone continues their ventilation throughout their illness and dies with it in place.

While a small number of studies have endeavoured to discover the experiences of the family members [13,38,50,51,52], the two studies which had some focus on patient views omitted the considerable advances made in home mechanical ventilation since their publication [37,47]. There has been little exploration of how dying, when ventilation is being used, is planned for, how it actually happens and what impact this has on family members. We have an insufficient knowledge or understanding about how, when or by whom ventilation interfaces should be removed once the patient has died, or the impacts of this potential physical barrier on the dying process. There is some recognition in the current literature that patients with MND may die suddenly, despite ventilation being in place, but there is no examination of how this happens or what impacts this might have [48]. While a large proportion of documents refer to and endorse the need for planning, this remains a complex and challenging issue for both health professionals and families, especially when ventilation is put in place in an emergency, and how this can been be implemented in practice requires our sustained attention. Qualitative studies exploring issues around dying with ventilation in place are needed. Particular emphasis needs to be placed on understanding the patient and family experience of ventilation use at the end of life. Larger, international studies to explore cross-cultural issues would also be of benefit.

### Strengths and Limitations of the Review

This scoping review comprehensively identified the types of written evidence available and mapped and summarised the evidence in order to establish what is known and identify the gaps in knowledge about the use of ventilation at the end of life in MND. The challenge with any literature review is to ensure that all the available literature has been identified and assessed. We recognise that we may not have included all grey literature, other than the UK guidance, which may not thus be inclusive of the diverse approaches in countries other than the UK. It should be noted that restrictions on the translation of non-English documents mean that we may not have been able to access a more substantial evidence base from Japan, where the withdrawal of ventilation is not an option for patients with MND, yet ventilation use is considerably higher than in other parts of the world. However, the nature of the topic, size of the population and the methodological limitations imposed by researching dying in MND mean that, in order to truly investigate the scope of the literature, it was vital to include case presentations, articles based on clinical experience and clinical guidance in this review.

It is also important to acknowledge that the use of ‘decision making’ and its derivatives in the searches is likely to have over-broadened the search, resulting in considerably more articles being identified than expected. This meant that it was necessary for E.W. to undertake initial screening of the titles only in order to narrow the records to a manageable size for more detailed examination. It is therefore possible that, with only one person involved in this phase, something could have been missed. In order to counteract this, we conducted extensive reference searching alongside a search of Google Scholar and the MedRxiv pre-print site. A further limitation to this review may be that we did not seek patient and public involvement. We further recognise the cultural specificity of elements such as decision making and the initiation of ventilation that may have limited our searching.

## Figures and Tables

**Figure 1 brainsci-12-01162-f001:**
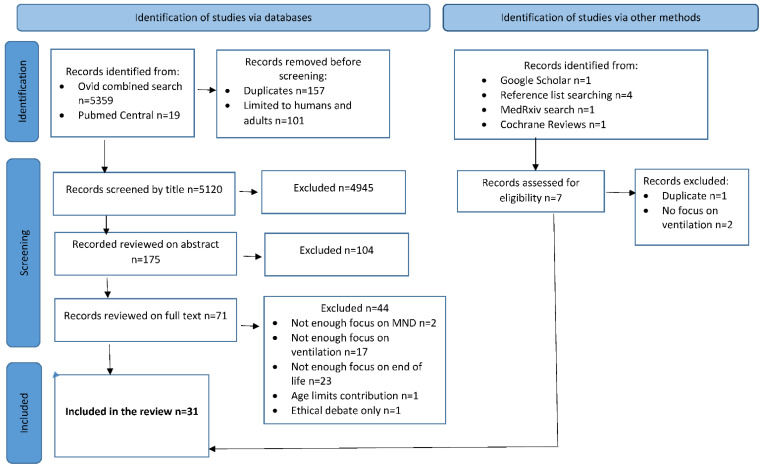
PRISMA flow chart of the literature screening process.

**Figure 2 brainsci-12-01162-f002:**
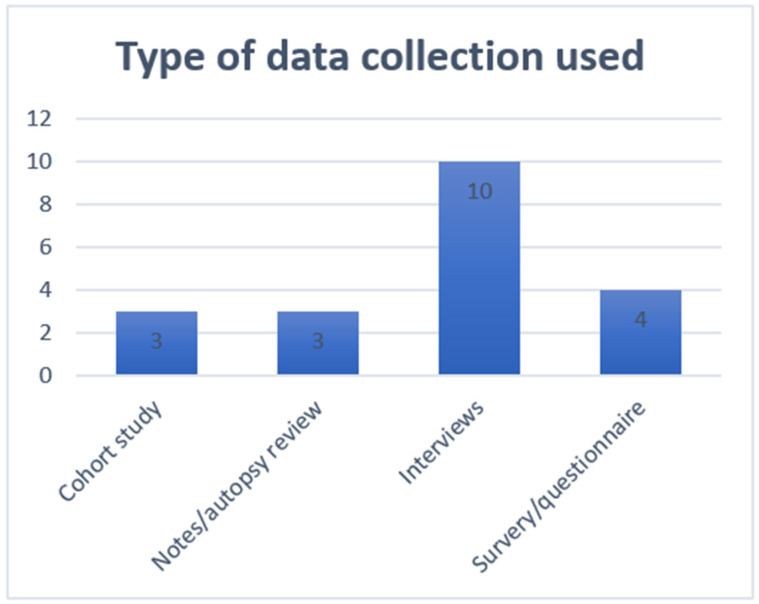
Study design.

**Figure 3 brainsci-12-01162-f003:**
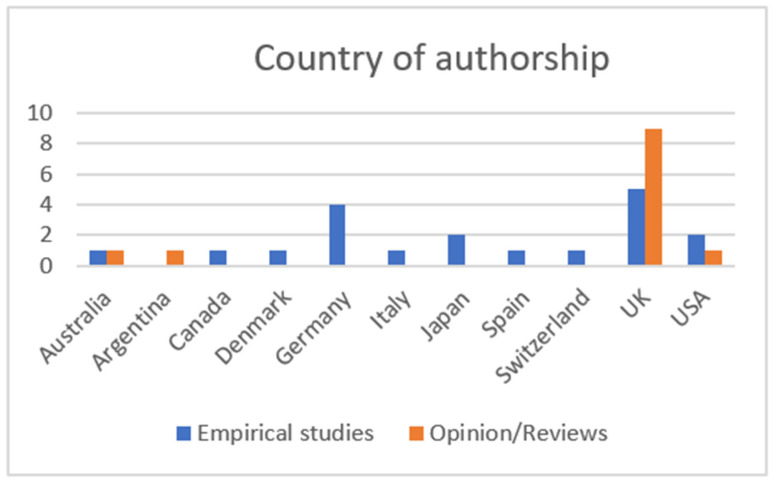
Country of authorship.

**Figure 4 brainsci-12-01162-f004:**
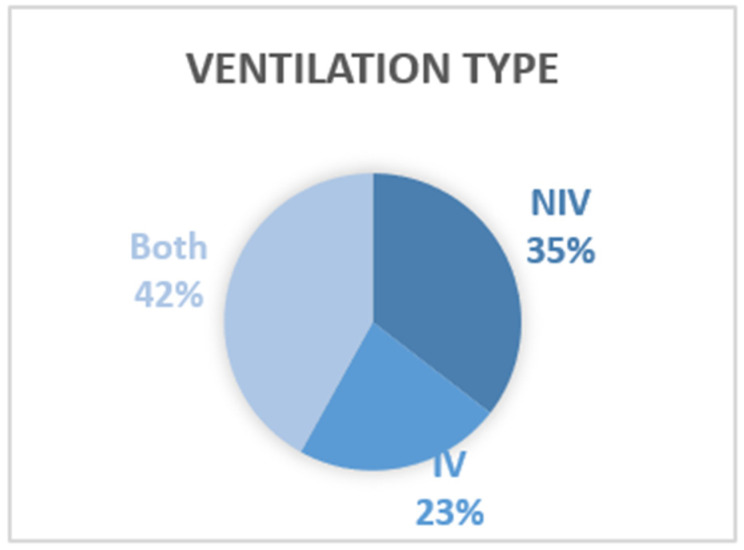
Ventilation type.

**Table 1 brainsci-12-01162-t001:** Inclusion criteria and sources of the evidence selection.

Criteria	Inclusion	Exclusion
Population	Patients with MND and experiences of using ventilation Family caregiver with experience of supporting someone with MND in using ventilation Health professional with experience of working with someone with MND using ventilation	Publications not including patients with MND/ALS
Concept	Use of ventilation Withdrawal of ventilation Dying with ventilation in place Non-invasive ventilation Invasive or tracheostomy ventilation Mechanics of ventilation Decision making Medications for withdrawal of ventilation/symptom management	Publications not involving ventilation at the end of life for patients with MND
Context	Health care settings Home Worldwide	
**Types of evidence sources**		
Study Design	Qualitative, quantitative, mixed methods, observational, experimental, clinical trials, quasi experimental studies, case studies, reviews	Letters and conference abstracts
Publication Type	Peer-reviewed publications, conference proceedings where a full report is available, textbook chapters, books, reports, preprint repositories, UK national guidance	Unable to obtain full text Local guidance provided by NHS Trusts
Language	English or those that can be translated into English using online translation tools	

**Table 2 brainsci-12-01162-t002:** Examples of variability in the reported place of death.

Paper	Country	Home/Nursing Home	Hospital
Burkhardt et al. [42]	Switzerland	60 (75%)	20 (25%)
Kuhnlein et al. [38]	Germany	18 (62%)	11 (38%)
Veronese et al. [50]	Italy	9 (53%)	8 (47%)
Ushikubo et al. [48]	Japan	4 (40%)	5 (50%) (1 in transit–10%)

## Data Availability

No data reported.

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
