# Peer review of "The Use of Mechanical Ventilation Support at the End of Life in Motor Neurone Disease/Amyotrophic Lateral Sclerosis: A Scoping Review"

_brainsci, 2022, doi:10.3390/brainsci12091162_

Round 1

Reviewer 1 Report

Comments and Suggestions for Authors

This is a very interesting paper with a nice summary in the beginning about the statistics around how many patients with MND are invasively ventilated, how that came about, legal issues around coming off ventilation, etc. From the manuscript, which focused on reviewing 31 papers discussing non-invasive ventilation, invasive ventilation, or both: Five themes were identified: Place of death, cause of death, comment on death, 182 planning, and ventilation withdrawal. Within the theme of ventilation withdrawal several 183 important subthemes were identified: timing and reason for withdrawal, ethical and emo- 184 tional issues, practical issues, including medication use, time until death and who should 185 be involved in the process.

1.  It would better separate out papers talking about non-invasive ventilation versus invasive ventilation, since the findings were clearly different with the different themes above, as summarized in the last sentence discussing place of death as one example.  There is a nice diagram listing which paper studied which types of patients, but it was tiring to keep flipping back to that figure when wanting to know which types of patients a particular paper was discussing.  For example the authors could say something like “For patients using non-invasive ventilatrion the following papers demonstrate an equal likelihood to die at home as in the hospital… and then share the data you site for these papers.  They could then move on to the next series of papers and say “for patients using invasive ventilaton, the following papers demonstrate a greater likelihood to die at home. . and then thare the datat you site for these papers”  This same comment applies to all of the themes and subthemes listed above.  This is better separated out in the time to death paragraph

2.  A figure or a comment about the current EOLOA laws in the country/state where the papers were published would be useful—perhaps this could be done in a sigmilar diagram to the papers focusing on NIV versus IV or both but intead having “no EOLOA, restricted EOLOA, open EOLOA” or something like that.

3.  Consider more figures to visually explain some of what is in the text since it is quite text heavy and also it might be easier to help with what I suggested in point 1

Reviewer 2 Report

Comments and Suggestions for Authors

Dear authors, in principle, I find the manuscript valuable. Here are some comments on the work, which should improve it.

Introduction:

1-45 Related to Invasive Ventilation and the not purely national focus of the article. I would suggest that some international examples (Japan, Germany, etc) be given here, or a range be given, especially since in the UK the number of invasively ventilated persons is relatively small. The rates might be up to 10 % even in Europe (https://erj.ersjournals.com/content/56/suppl_64/3419 doi: 10.4045/tidsskr.20.1030

https://pubmed.ncbi.nlm.nih.gov/33210770/)

2-50 I would recommend avoiding repeatedly emphasizing that discontinuing ventilation that has already been initiated is a difficult decision. In principle, this is self-evident when the patient dies. The complexity of the decision-making process may start much earlier than when the patient is already ventilated. Advanced Care Plannig should already include decisions for or against ventilation before the initiation of any ventilatory measure. Situations in which ventilation should not be continued should also be defined (e.g., loss of all communication). If the ACP has been done appropriately, the decision may lose some of its difficulty. Here, the chapter on "Planning" that appears later in the manuscript could be somewhat anticipated.

Methods:

2/83 JBI should be written out again here.

05/176 Table 2: To make the table clearer, the header could be included on the following pages.

Findings: Figure 2: The figures should be revised. The authors should manage to make the essential information clear at first look. The Y axis should not be omitted (number). The description of the columns should not be abbreviated by dots. If necessary, write the words one below the other.

06/212 Figure 3 could also stand for itself without concluding that the UK leads the science field. The very fact, that only one non-English publication was included, which I appreciate, but no national guidelines from non-English speaking countries were considered, I would recommend a little more restraint. Certainly, the number of publications is considerable, but I would formulate the conclusion less explicitly.

Discussion 11/448

Of course, it is interesting why there is little evidence from Japan on this topic, as well as from all other countries where more invasive ventilation is performed than in the UK. However, there is work from Japan on this: https://doi.org/10.1159/000341341

Because of the cultural peculiarities in Japan that can be found in the literature, I would not be so critical of Japan here. DOI: 10.5772/intechopen.69773  And I must mention it again, some Japanese literature will certainly not have found any attention in this manuscript.

From personal conversations with Japanese colleagues, I must emphasize that the topic is certainly not something that is not of interest in Japan. Therefore, this work can be very important for Japan and the rest of the world, if it is culturally better classified. I would refrain from highlighting the British guideline too much. From my own experience, I can only say that guidelines can be very substantial in other countries as well.

I recommend mentioning the cultural specificity of decision-making and initiation of ventilation in the limitations. Even if the limited human resource has already been mentioned.

Even though this is a review paper, I would like to see it contribute in the discussion of improving mechanical ventilatory support at the end of life in ALS patients through good advanced care planning, as I see this as a key role.

While this was a brief mention in point 6, I think it is very relevant and could be addressed.

Reviewer 3 Report

Comments and Suggestions for Authors

This scoping review deals with the topic of mechanical ventilation support at the end of life in ALS, a relevant topic for clinicians involved in the care for ALS patients, such as neurologists or palliative care physicians. The analysis is based on 31 documents including scientific studies, clinical case examples and guidelines. The review is carefully executed and the data gathered from the literature is well documented.

The authors identify several areas of information that are presented in these studies, such as place and cause of death and advance planning and highlight the topic of withdrawal of ventilation, which several studies focus on. The authors conclude from their analysis, that little is known about dying with mechanical ventilation in place and experiences and views of patients and caregivers.

In my opinion, this scoping review deserves publication. I would, however, recommend to add some information to the introduction part as to how often non-invasive ventilation is introduced in ALS patients in European countries. In the discussion section, I would recommend to add some thoughts about why there is a lack of information about the end of life of ALS patients. Are palliative care teams or neurologists only involved, if the patient asks for withdrawal? What conclusions do the authors draw from the identified lack of knowledge – what types of studies are needed?

Round 2

Reviewer 1 Report

Comments and Suggestions for Authors Yes, I feel the authors have addressed the reviewer comments and modified the manuscript accordingly whenever possible. I support this paper being published now.

Author Response

Thank you for your response